# Water-Soluble Fiber from Bengkoang (*Pachyrhizus erosus* (L.) Urban) Tuber Modulates Immune System Activity in Male Mice

**Hanif Nasiatul Baroroh** [1,2], **Agung Endro Nugroho** [1], **Endang Lukitaningsih** [3] **and Arief Nurrochmad** [1,*]

1   Department of Pharmacology and Clinical Pharmacy, Faculty of Pharmacy, Universitas Gadjah Mada, Sekip Utara Yogyakarta 55281, Indonesia; baroroh2015@gmail.com (H.N.B.); nugroho_ae@ugm.ac.id (A.E.N.)
2   Department of Pharmacy, Faculty of Health Sciences, Jenderal Soedirman University, Purwokerto 53122, Indonesia
3   Department of Pharmaceutical Chemistry, Faculty of Pharmacy, Universitas Gadjah Mada, Sekip Utara Yogyakarta 55281, Indonesia; lukitaningsih_end@ugm.ac.id
*   Correspondence: ariefnr@ugm.ac.id; Tel.: +62-0274-543120

**Abstract:** Our previous study showed that water-soluble fiber from bengkoang (*Pachyrizus erosus* (L.) Urban) fiber extract (BFE) and bengkoang fiber fraction B (BFE-B) have phagocytic activity and modulation of cytokine production in vitro. The present study evaluates the immunomodulatory effects of water-soluble fibers BFE and BFE-B on male mice induced by hepatitis B vaccine. Thirty mice were divided into six groups and induced by hepatitis B vaccine intraperitoneally on days 7 and 14. The mice were then treated with BFE, BFE-B, levamisole, or sodium carboxymethyl cellulose for 18 days. At the end of the treatments (day 19), phagocytic activity, lymphocyte proliferation, spleen index, cytokine, and immunoglobulin G (IgG) production were determined. The results showed that the water-soluble fiber treatment could significantly increase phagocytic capacity, nitric oxide production, and spleen index. However, BFE-B could modulate tumor necrosis factor (TNF)-$\alpha$ and interleukin (IL)-10 secretion, BFE demonstrated no such effect on cytokine production. Lymphocyte proliferation assay revealed that treatment with 50 mg/kg body weight (BW) BFE and 50 mg/kg BW BFE-B could significantly enhance lymphocyte proliferation. Treatment with 25 and 50 mg/kg BW BFE-B stimulated IgG production. In conclusion, BFE and BFE-B similarly have immunomodulatory effects on innate immune responses. BFE-B further demonstrated immunomodulatory effects on adaptive immune responses.

**Keywords:** immunomodulatory effect; *Pachyrhizus erosus* (L.) urban; water-soluble fiber; phagocytic activity; adaptive immune

## 1. Introduction

The immune system of the body protects and repairs cells from disease, especially infectious and chronic diseases. Modulation of the immune response requires an immunomodulator that can efficiently affect the humoral and cellular immune systems. The immune system can be divided into the innate immune system and the adaptive immune system [1]. Drugs often used as immunomodulators include immunostimulant and immunosuppressant drugs [2]. Alternative herbal medicine can be used to modulate and activate an immune response. Various herbal ingredients exhibit immunostimulatory activity. Indeed, several studies on the modulation and activation of the immune response by food containing dietary fiber have been published [3,4].



Herbal medicine that may be developed as an immunomodulatory agent is bengkoang (*Pachyrhizus erosus*), which has been proven to have antioxidant activity. Bengkoang contains nutrients and chemical compounds that could potentially be developed as functional food and phytotherapy. Bengkoang contains vitamin C, vitamin B1, protein, and a relatively large amount of dietary fiber. Bengkoang has a low caloric value of 39 kcal/100 g because it contains inulin [5]. Noman et al. [5] reported that bengkoang contains 14.9% carbohydrates and 1.4% fiber; the herb also contains isoflavonoid compounds such as daidzein and furanyl pterocarpan, which are believed to be potential antioxidant sources [6]. According to Ramirez-Santiago et al. [7], bengkoang contains pectin and hemicellulose polysaccharides, both of which show potential immunomodulatory activity.

Several investigations on the immunomodulatory activity of bengkoang fiber extract (BFE) have been reported. BFE could enhance immune system through the mitogen-activated protein kinase (MAPK) and nuclear factor (NF)-κB pathways [8], enhance the phagocytic activity of macrophages, and stimulate the production of TNF and IL-6 on murine macrophage-like J774.1 cells [9]. BFE exerts immunomodulatory effects in vitro and in vivo by increasing immunoglobulin G (IgG), IgA, and IgM levels and IL-5 and IL-10 production [10]. Our previous study on water-soluble fiber and some of its fractions from bengkoang (*P. erosus*) showed that BFE and BFE-B contained pectin-like compound and could enhance phagocytic activity and modulate cytokines production in vitro [11]. However, to date, no report on the immunomodulatory effect of the water-soluble fiber fraction of *P. erosus* in mice induced by hepatitis B vaccine is yet available. Therefore, in the present study, the water-soluble fiber fraction of bengkoang was purified to increase the selectivity of its active compound as an immunomodulator to develop the applications of natural herbs further. The present study aimed to determine the immunomodulatory effects of water-soluble fiber, BFE and BFE-B on mice induced by hepatitis B vaccine.

## 2. Materials and Methods

### 2.1. Materials

Ethanol, ammonium oxalate, potassium hydroxide, sodium carboxymethyl cellulose (CMC-Na), sulfanilamide, phosphoric acid, *N*-(1-naphthyl) ethylenediamine dihydrochloride, tetrazolium salt (3-(4,5-dimethylthiazol-2-yl)-2,5-diphenyl tetrazolium bromide (MTT)) and latex beads were purchased from Sigma–Aldrich Pte. Ltd., Singapore. Hepatitis B vaccine (Engerix B) was obtained from Glaxo SmithKline, and levamisole was obtained from PT Konimex, Indonesia. Rosewell Park Memorial Institute (RPMI) 1640 medium, 2% penicillin–streptomycin, 0.5% fungizone, 10% fetal bovine serum, and phosphate-buffered saline (PBS) were purchased from Life Technologies Corporation (GIBCO). Mouse IL-10, TNF-α, and IgG enzyme-linked immunosorbent assay (ELISA) kits were purchased from Fine Test, Wuhan Fine Biotech Co., Ltd., Wuhan, China. All other reagents and chemicals used in this work were of the purest commercial grade available.

### 2.2. Plant Material Collection

*P. erosus* tubers aged approximately five months were obtained from Prembun, Kebumen, Central Java, Indonesia. The plant was identified and authenticated by Dr. Djoko Santosa, an Assistant Professor at the Department of Pharmaceutical Biology, Faculty of Pharmacy, Universitas Gadjah Mada, Yogyakarta, Indonesia.

### 2.3. Animals

Male BALB/c mice, 6–8 weeks old and weighing approximately 25–30 g, were obtained from the Animal Center of Pharmacology and Toxicology Laboratory, Universitas Gadjah Mada. All animals were acclimatized for one week before experimentation and housed in an animal room with a temperature of (22 ± 3) °C and humidity of 60% under a 12-h light/dark cycle. A standard pelleted basal diet and water were provided ad libitum. Animals were randomized into experimental and control groups.

All experimental procedures were approved by the Institutional Animal Ethics Committee of the Integrated Research and Testing Laboratory, Universitas Gadjah Mada (No. 00073/04/LPPT/VII/2018).

### 2.4. Preparation of the Soluble Fibers of P. erosus Tubers

The *P. erosus* tubers were peeled, mashed and dissolved in distilled water (1:2) overnight, and then centrifuged to separate the starch and fiber (Figure 1). The precipitate was separated, and the supernatant was collected for further treatment. The supernatant was evaporated in a water bath for 30 min, immersed in 80% ethanol (1:1), filtered, and evaporated at 60 °C for 20 min. After filtration, the precipitate was freeze-dried until a dry powder was obtained. This product was designated BFE [9]. Exactly 100 g of BFE was added with 0.5% (*w/v*) ammonium oxalate solution (1:2), heated in boiling water bath for 30 min, centrifuged at 1400× *g* for 5 min, and then filtered. Cold ethanol (−20 °C, 4×) was mixed with the supernatant at (5 ± 2) °C for 12 h. The mixture was subsequently centrifuged at 1400× *g* for 5 min, and the precipitate was freeze-dried. The resulting fiber fraction was designated BFE-B [7,12].

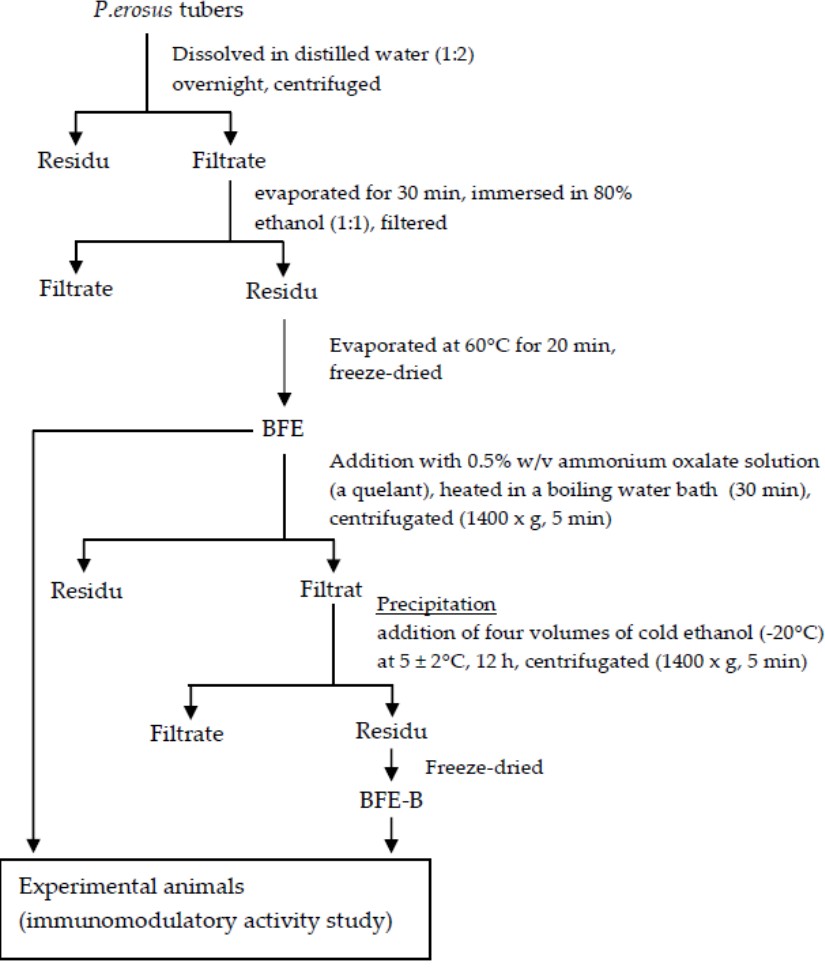

**Figure 1.** Preparation of water-soluble fibers of *P. erosus* tubers.

### 2.5. Experimental Animals

In vivo immunomodulatory activity was evaluated using the posttest-only control group design method with a randomized directional pattern design (Figure 2). Thirty male mice were randomly divided into six groups, each consisting of five animals. All of the mice were injected with hepatitis B vaccine (20 µg/mL) intraperitoneally (i.p.) at a volume of 2.6 µL/20 g BW on days 7 and 14 [13]. The mice were then treated orally for 18 days. Group 1 was given 0.5% CMC-Na, group 2 was given

2.5 mg/kg BW levamisole, groups 3 and 4 were, respectively, given 50 and 100 mg/kg BW BFE, and groups 5 and 6 were respectively given 25 and 50 mg/kg BW BFE-B. In the present study, levamisole was used as a positive control. Levamisole could stimulate phagocytic capacity, increase NO production by macrophages, and induce lymphocyte proliferation. Thus, levamisole is a drug that could modulate cellular immune responses. On day 19, the mice were anesthetized with 180 mg/kg BW ketamine intraperitoneally (i.p.) for blood sampling. Mice were then sacrificed and peritoneal macrophages and spleen were taken out. Assessments of spleen index, phagocytic activity, nitric oxide (NO) production, lymphocyte proliferation, and TNF-α, IL-10, and IgG production were then conducted.

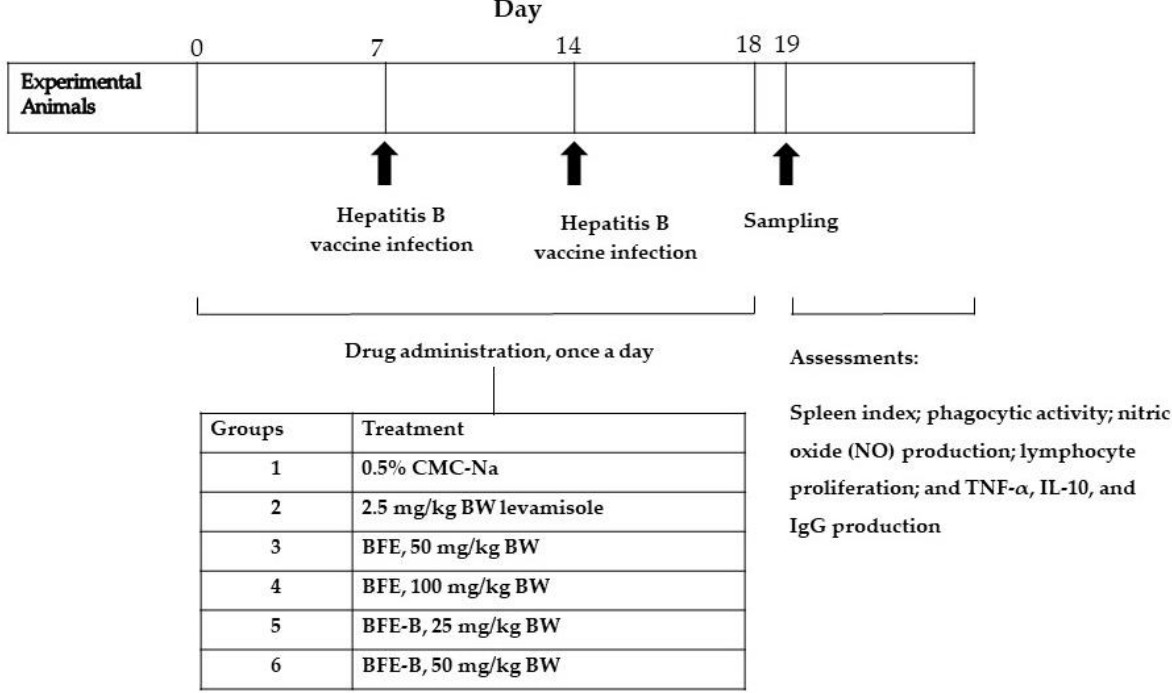

**Figure 2.** Experimental animals design for experimental analysis.

## 2.6. Phagocytic Assay of Macrophages

The skin of the abdomen was cut open, and the peritoneal cavity was injected with 10 mL of RPMI. The peritoneal fluid containing macrophages was then removed, centrifuged at 1500 rpm and 4 °C for 10 min, and resuspended in 5 mL of complete medium. A cell suspension with a density of $2.5 \times 10^6$ cells/mL was cultured in 24-well plates fitted with spherical coverslips. Each well contained 200 μL ($5 \times 10^5$ cells) of the suspension and was incubated in a $CO_2$ incubator at 37 °C overnight. Thereafter, the medium in each well was collected for further treatment (NO production assay) and incubated in a $CO_2$ incubator at 37 °C for 4 h. After incubation, a suspension of 200 μL of latex beads ($2.5 \times 10^7$ beads/mL) was placed in each well and then incubated in a $CO_2$ incubator at 37 °C for 60 min. The cells were washed with PBS (3×), dried, and fixed with methanol for 30 s. After drying, the coverslips were painted with 20% (*v/v*) Giemsa reagent for 30 min, washed with distilled water, and then dried once more. The number of macrophages on the slides was observed with a microscope at 400× magnification. One hundred macrophage cells were observed, and the numbers of macrophages that successfully phagocyted the latex particles (as a measure of phagocytic capacity) and latex beads that were phagocyted by macrophages (as a measure of phagocytic index) were counted [14].

## 2.7. Nitric Oxide (NO) Production Assay

The NO production is measured by measuring nitrite concentrations in macrophages cell supernatants using Griess reagents. The standard nitric oxide (NO) solutions were prepared by

dissolving 69.0 mg of sodium nitrite in 100 mL aquabidest to produce a 2000 μM standard NO solution and then diluted to obtain a series of standard NO solutions with concentrations ranging from 0 μM to 100 μM. A total of 100 μL of each macrophage cell culture that had been incubated overnight was placed in wells and then 100 μL of Griess reagent (1% sulfanilamide in 5% phosphoric and 0.1% *N*-(1-naphthyl) ethylenediamine dihydrochloride in water) was added. The mixtures were then incubated at room temperature for 15 min, and the absorbance of each well was determined at 550 nm using a microplate reader. The amount of nitrite in the sample was calculated by plotting to a standard curve of sodium nitrite concentration (0–100 μM).

### 2.8. Spleen Index Analysis

On day 19, after the final administration of water-soluble fibers, mice were sacrificed, and the spleen was dissected and weighed. The spleen index was then determined as follows:

$$\text{Spleen index} = \frac{\text{Spleen weight (mg)}}{\text{body weight of mice (g)}} \times 10 \tag{1}$$

### 2.9. Lymphocyte Proliferation Assay

As much as 10 mL of RPMI was injected into the spleens of the mice until all of the lymphocytes were completely extracted. The suspension was placed in a microtube tube and centrifuged for 10 min at 1500 rpm and 4 °C. The pellets were hemolyzed two times using a lysis buffer and the centrifuge at 3000 rpm and 4 °C for 4 min. The supernatant was removed, and the precipitant was collected so that it contained the lymphocytes inoculated into the RPMI 1640 medium, penicillin-streptomycin 2%, fungizone/amphotericin B (GIBCO) 0.5%, and fetal bovine serum (GIBCO) 10%. A suspension of lymphocyte cells with a density of $1.5 \times 10^6$ cells/mL was cultured in a 96-multiwell plate and incubated in a $CO_2$ incubator at 37 °C for 48 h. Cell viability was measured by tetrazolium salt (MTT) assay, and the optical density (OD) of each well was determined using a microplate reader at a wavelength of 550 nm.

### 2.10. Production of TNF-α, IL-10, and IgG

Blood samples were collected through the retro-orbital plexus into capillary tubes and then allowed to stand for 2 h at room temperature. Blood was centrifuged at 4000 rpm at 25 °C for 20 min to obtain the serum. Determinations of TNF-α, IL-10, and IgG levels in the serum were carried out using the corresponding mouse ELISA kits according to manufacture protocol.

### 2.11. Statistical Analysis

Data were presented as mean ± standard error of the mean (SEM), and statistically significant differences between groups was analyzed by one-way analysis of variance or the Kruskal–Wallis test followed by the Least Significant Difference (LSD) or Mann–Whitney test. Statistical analysis was performed using SPSS version 25, and $p < 0.05$, $p < 0.01$, and $p < 0.001$ were considered to indicate statistically significant differences.

## 3. Results

### 3.1. Phagocytic Activity of Macrophages

The activity of phagocytic macrophages against latex was evaluated in terms of the number of activated macrophages (phagocytic capacity) and the activity of each macrophage unit (phagocytic index). Administration of water-soluble fiber increased the number of activated macrophages in mice and increased improved the capacity of macrophages to phagocytize latex after induction by hepatitis B vaccine.

As shown in Figure 3, mice were treated with 50 mg/kg BW BFE and 50 mg/kg BW BFE-B increased the phagocytic index of macrophages to 3.59 and 3.54, respectively ($p < 0.001$). No significant increase in phagocytic index was noted after treatment with levamisole or 100 mg/kg BW BFE. In the observation of phagocytic capacity, levamisole, BFE, and BFE-B could enhance phagocytic capacity ($p < 0.001$). Among the treatments investigated, 100 mg/kg BW BFE showed the greatest ability to stimulate phagocytic capacity, which was 54.6% higher than that obtained by treatment with levamisole (Table 1). Dose-dependent improvements in phagocytic capacity were observed after the administration of *P. erosus* water-soluble fiber. This result shows that *P. erosus* water-soluble fiber could stimulate non-specific immune responses.

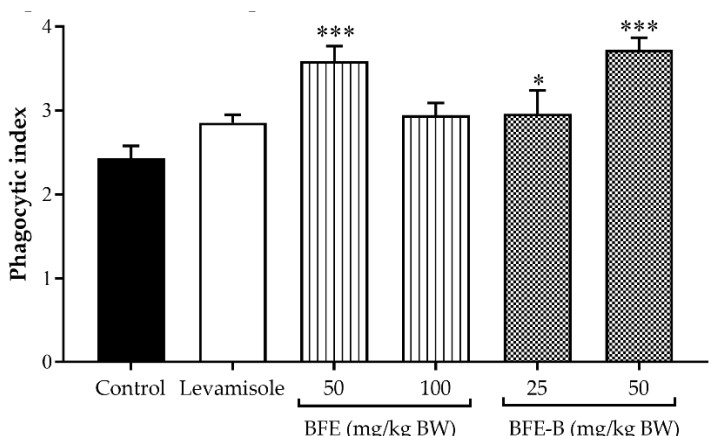

**Figure 3.** Effect of water-soluble bengkoang (*P. erosus*) fiber extract (BFE) and fiber fraction B (BFE-B) on the phagocytic index of peritoneal macrophages of mice induced by hepatitis B vaccine. The data represent the mean ± SEM of five independent measurements. Significant differences compared with the control are determined at * $p < 0.05$ and *** $p < 0.001$.

**Table 1.** Effect of water-soluble fibers of *P. erosus* (BFE and BFE-B) on the phagocytic capacity of peritoneal macrophages of mice induced by hepatitis B vaccine.

| Group | Phagocytic Capacity (%) ± SEM |
|---|---|
| BFE, 50 mg/kg BW | 49.60 ± 7.12 *** |
| BFE, 100 mg/kg BW | 54.60 ± 6.21 *** |
| BFE-B, 25 mg/kg BW | 48.40 ± 2.13 *** |
| BFE-B 50 mg/kg BW | 50.50 ± 3.26 *** |
| Levamisole, 2.5 mg/kg BW | 53.75 ± 2.93 *** |
| Control (CMC-Na) | 22.20 ± 5.80 |

The data represent the mean ± SEM of five independent measurements. Significant differences compared with the control are determined at *** $p < 0.001$.

## 3.2. Nitric Oxide Production in Mouse Peritoneal Macrophage

NO was produced and released from peritoneal macrophages to the medium after incubation for 24 h. The NO concentrations detected in each group are presented in Table 2.

The administration of BFE showed a tendency to increase NO production, but not significantly different to control. However, administration of BFE-B (25 and 50 mg/kg BW) and levamisole could activate NO production and significantly increased NO concentrations by 7.4 ($p < 0.001$; $p < 0.01$) and 4.5-fold ($p < 0.05$) compared to the control, respectively (Table 2). These results also demonstrate the dose-dependent effect of BFE-B. NO production in mice treated with BFE-B was higher than that in mice treated with BFE and levamisole.

**Table 2.** Effect of water-soluble fibers of *P. erosus* (BFE and BFE-B) on nitric oxide production in the peritoneal macrophages of mice induced by hepatitis B vaccine.

| Group | Nitric Oxide Concentration ($\mu$M) $\pm$ SEM |
|---|---|
| BFE, 50 mg/kg BW | $1.77 \pm 0.26$ |
| BFE, 100 mg/kg BW | $0.821 \pm 0.04$ |
| BFE-B, 25 mg/kg BW | $4.199 \pm 0.36$ *** |
| BFE-B, 50 mg/kg BW | $3.47 \pm 0.98$ ** |
| Levamisole, 2.5 mg/kg BW | $2.59 \pm 0.06$ * |
| Control (CMC-Na) | $0.57 \pm 0.02$ |

The data represent the mean $\pm$ SEM of three independent measurements. Significant differences compared with the control are determined at * $p < 0.05$, ** $p < 0.01$, and *** $p < 0.001$.

### 3.3. Spleen Index

In this study, the spleen index was determined by weighing the spleen and comparing it with the body weight of the mice. The spleen indices of different groups are illustrated in Figure 4. The spleen index increased in all treatment groups. Among the treatments studied, the treatment with 25 mg/kg BW BFE-B resulted in the highest increase in spleen index, and differences between this group and the control were significant ($p < 0.001$). This result shows that the administration of water-soluble fiber for 18 days could increase the immune response of mice induced by hepatitis B vaccine.

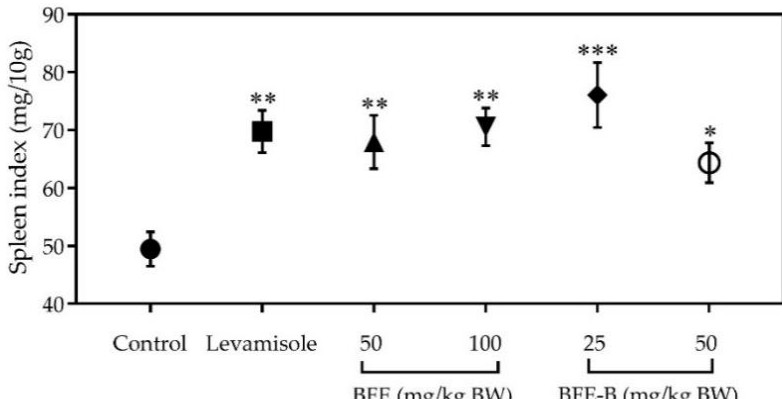

**Figure 4.** Effect of water-soluble fibers of *P. erosus* (BFE and BFE-B) on the spleen index of mice induced by hepatitis B vaccine. The data represent the mean $\pm$ SEM of five independent measurements. Significant differences compared with the control are determined at * $p < 0.05$, ** $p < 0.01$, and *** $p < 0.001$.

### 3.4. Lymphocyte Cell Proliferation

Lymphocyte cell proliferation assay was carried out to determine the effects of water-soluble fibers of *P. erosus* on specific immune responses. Here, lymphocyte proliferation was determined by comparing the OD of each treatment group with that of the control.

The increase in OD in all treatment groups compared with that in the control group indicates an increase in lymphocyte cell proliferation. Administration of levamisole, BFE, and BFE-B at a dose of 50 mg/kg BW increased lymphocyte cell proliferation ($p < 0.001$), as indicated in Figure 5. Among the treatments studied, the treatment of BFE-B, 50 mg/kg BW revealed the greatest immunostimulatory effect on lymphocyte proliferation, increasing lymphocyte cell production by 83.06%. This result indicates that the water-soluble fiber fraction of *P. erosus* could modulate the adaptive immune response of mice.

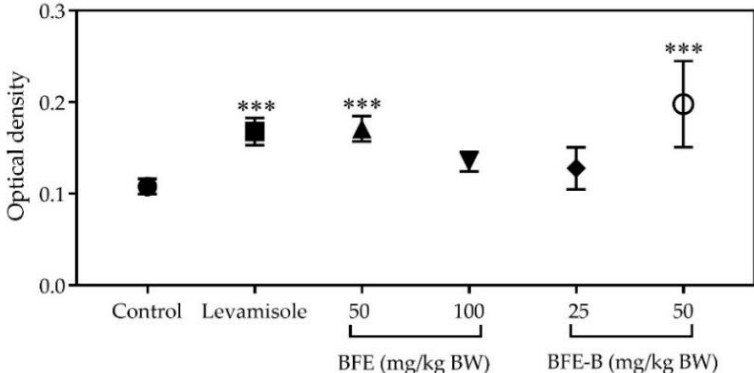

**Figure 5.** Effect of water-soluble fibers of *P. erosus* (BFE and BFE-B) on the proliferation of lymphocytes in mice induced with hepatitis B vaccine. Lymphocyte cells were cultured for 48 h. After cultivation, cell viability was measured by tetrazolium salt (MTT) cell proliferation assay and the absorbance was read at 550 nm. The data represent the mean ± SEM of five independent measurements. Significant differences compared with the control are determined at *** $p < 0.001$.

### 3.5. Cytokines Production

The result showed that administration of BFE (50 and 100 mg/kg BW) did not increase TNF-α production, whereas the administration of BFE-B, 25 mg/kg BW increased TNF-α production in mice induced by hepatitis B vaccine ($p < 0.001$) (Figure 6A). Treatment with 50 mg/kg BW BFE-B increased TNF-α production, but the levels observed were not significantly different compared with the control (Figure 6A). While, administration of BFE-B, 50 mg/kg BW significantly increased the production of IL-10 ($p < 0.05$), but levamisole, BFE (50 and 100 mg/kg BW), and BFE-B, 25 mg/kg BW exerted no such effect on IL-10 levels in mice induced by hepatitis B vaccine (Figure 6B).

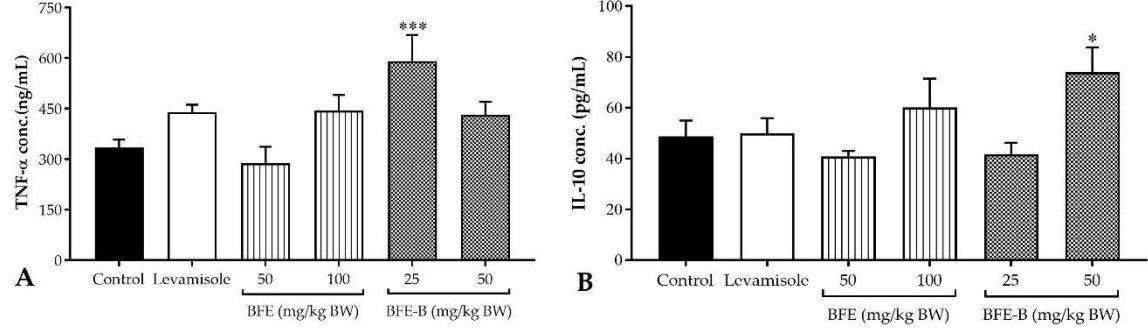

**Figure 6.** Effect of water-soluble fibers of *P. erosus* (BFE and BFE-B) on TNF-α (**A**) and IL-10 (**B**) production in mice induced by hepatitis B vaccine. The data represent the mean ± SEM of five independent measurements. Significant differences compared with the control are determined at * $p < 0.05$ and *** $p < 0.001$.

### 3.6. Immunoglobulin G Production

As shown in Figure 7, increased IgG levels were observed after administration of 2.5 mg/kg BW levamisole, but no significant difference compared with the control was found. BFE treatments of 50 and 100 mg/kg BW showed no effect on IgG levels. However, the IgG titer significantly increased after treatment with 25 and 50 mg/kg BW BFE-B ($p < 0.05$).

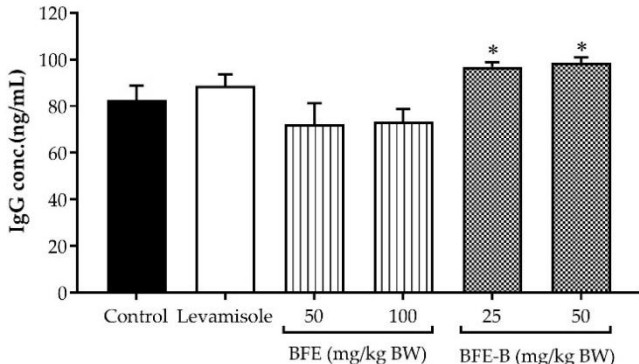

**Figure 7.** Effect of water-soluble fibers of *P. erosus* (BFE and BFE-B) on IgG production in mice induced by hepatitis B vaccine. The data represent the mean ± SEM of five independent measurements. Significant differences compared with the control are determined at * $p < 0.05$.

## 4. Discussion

Alternative herbal medicine can be used to modulate and activate immune responses, and plants with immunomodulatory activity are often used as adjuvant therapy in chemotherapy. One plant that could be considered an adjuvant therapy is bengkoang (*P. erosus*). Bengkoang contains water-soluble fiber, which has potential use as an immunomodulator. In previous studies, isoflavone compounds from *P. erosus* were proven to inhibit the proliferation of estrogen-dependent breast cancer (T47D) cells in vitro [15]. In the present study, water-soluble from bengkoang enhanced the immune system; thus, it may be further developed as adjuvant therapy.

Several lines of evidence have reported that polysaccharide and fiber compounds could modulate immune responses. (1,3)-$\beta$-D-Glucan, for instance, demonstrates immunostimulatory activity [16]. Huang et al. [17] investigated the ratio of mannose, the backbone of the $\beta$-(1→4)-Man linkage, and found that a large proportion of *O*-acetyl groups contributes to the immunomodulatory activity of *O*-acetyl-glucomannan (Dendronan). Some active compounds of polysaccharides showing immunomodulatory activity in mushroom include $\beta$-D-glucan, galactomannan, lentinan, and acidic polysaccharides [18,19]. The soluble fiber of *Moringa oleifera* seeds demonstrates immunomodulatory activity by enhancing lymphocyte proliferation and NO production [20]. Homogalacturonan pectin has also been reported to increase macrophage activity and NO production [21]. Differences in immunomodulatory activity may be influenced by the compounds that make up different types of water-soluble fiber. BFE is a crude fiber that contains pectin and other polysaccharide compounds that are likely to affect the immunomodulatory effect. In this study, we demonstrate for the first time that the fiber fraction from *P erosus*, BFE-B, which contained pectin-like compound can modulate immune system in mice.

The water-soluble fiber of *P. erosus* could modulate the specific and non-specific responses of the immune system of mice induced by hepatitis B vaccine. BFE could increase phagocytic activity, NO production, and lymphocyte proliferation, but does not affect IgG and cytokines (TNF-$\alpha$ and IL-10) production. Similarly, BFE-B stimulated macrophages to carry out phagocytosis, lymphocyte proliferation, and NO, TNF-$\alpha$, IL-10, and IgG production. In our previous study, the effect of water-soluble fiber fraction of *P. erosus* on cytokine production has been studied in vitro. The results indicated that the BFE-B enhances TNF-$\alpha$ and IL-6 production but inhibits IL-10 production [11]. In line with the previous report, oral treatment with high doses of the crude fiber of *P. erosus* could significantly stimulate IL-6, IL-10, TNF-$\alpha$, transforming growth factor (TGF)-$\beta$, and interferon (IFN)-$\gamma$ production by lymphocytes from the spleen [10]. The crude fiber of bengkoang could enhance TNF-$\alpha$ and IL-6 production in J774.1 cells and P-mac ex vivo when applied at a dose of 27 mg/kg BW [9]. Levamisole enhances lymphocyte proliferation activity and interferon production and stimulates dendritic cell maturity [22]. Levamisole has been reported to stimulate IL-12, IL-10, INF-$\gamma$, IL-5, and

TNF-$\alpha$ production [23,24]. In the present study, however, levamisole showed no effect on TNF-$\alpha$, IL-10, and IgG production.

In this study, water-soluble fiber of *P. erosus* also enhanced the phagocytic activity and NO production of macrophages. Macrophages produce NO, which is involved in the regulation of apoptosis and defense against infection from microorganisms. NO secretion is a parameter that indicates increased macrophage activity. BFE-B could stimulate the production of pro-inflammatory (TNF-$\alpha$) and anti-inflammatory (IL-10) cytokines in macrophages. IL-10 is a cytokine produced by T helper (Th2) cells that suppresses the production of Th1 cells and inhibits macrophage activation. IL-10 is a cytokine that inhibits CD4+ T cell proliferation by downregulating major histocompatibility complex (MHC) II and increases the proliferation of CD8+ T cells [25]. Blockade of IL-10 increased the proportion of activated CD4+ and $\gamma\delta$ T cells and IFN-$\gamma$ production by CD4+ in response to antigen presentation, which, in turn, inhibits the protective immune response against secondary infections [26]. TNF-$\alpha$ enhances the ability of macrophages to carry out phagocytosis. TNF-$\alpha$ can modulate the immune response by inhibiting T cell receptor signaling, promoting apoptosis of lymphoid T cells, inhibiting dendritic cell (DC) co-stimulation, and stimulating other cytokines that can inhibit cell-mediated immunity [27]. The modulation of cytokine production by macrophages induced by BFE-B provides scientific evidence that the water-soluble fiber examined in this work has immunomodulatory activity.

Administration of BFE-B significantly increased IgG production on day 19 after the mice were induced by hepatitis B vaccine on days 7 and 14. A previous study revealed that oral administration of BFE doses of 6.75 and 27 mg/kg BW enhance IgG production in mice [10]. IgG levels were measured on day 19 because the level of this antibody peaks on days 10–14 after an antigen enters the body. A lag phase occurs on days 1–3 after repeated antigen infection, and IgG levels show a greater increase and longer effects during this period [1]. IgG is the main immunoglobulin formed by antigen stimulation [28]. The antibody coats microorganisms so that particles are easily phagocytosed, thereby neutralizing toxins and viruses. IgG has a half-life of approximately 23 days in the blood [29]. The lymphocyte cells that play a role in the production of immunoglobulins are Th2 and Treg cells. Th2 cells produce IL-4, which stimulates IgG production, while Treg cells could reduce antibody production by B cells [1].

The cellular mechanism of action of the water-soluble fiber of *P. erosus* as an immunomodulator remains unclear because different fiber extract or fractions show different activities in the innate and adaptive immune systems. Previously, our study reported that BEF contained dietary fiber and BEF-B was chemically characterized as containing a pectin-like compound [11]. Several studies reported the effect of fiber on immune system. Dietary fiber affects gut-associated lymphoid tissue, which plays a role in the immune system in the gastrointestinal tract [30]. Fiber fermentation helps stimulate the immune system in the large intestine [31,32]. Bacterial fermentation promotes the conversion of pectin into butyrate, propionate, and acetate [33]. Short chain fatty acids (SCFAs) induces an anti-inflammatory effect on the intestine by decreasing the intestine permeability index and toll-like receptor (TLR)-4, IL-6, and TNF-$\alpha$ expression. Butyrate can decrease leukocyte infiltration, but increase T cell activity, DCs in Peyer's patch, and IL-10 production [34]. Crude fiber-containing pectin-like compounds could activate macrophages through TLR4 dependent signaling pathways [8]. According to the present results, water-soluble fiber from bengkoang, BFE-B, could enhance the immune system and may be developed as adjuvant therapy. To explore the cellular mechanism of this potential immunomodulatory effect is recommended in future studies.

## 5. Conclusions

In our study, we provided evidence that the water-soluble fiber of *P. erosus*, BFE and BFE-B exerts immunomodulatory effects. We reported for the first time that water-soluble fiber fraction from *P. erosus*, BFE-B, stimulated phagocytotic macrophages, NO production from peritoneal macrophages, and lymphocyte proliferation in vivo. The increased phagocytic activity of macrophages was evidenced by enhancements in phagocytic capacity and index, which are characterized by increased NO, IL-10,

and TNF-α production. Increased lymphocyte proliferation triggers an increase in IgG production in mice after induction by hepatitis B vaccine. All our findings suggest that oral administration of water-soluble fiber from bengkoang, BFE, and BFE-B, could modulate the dynamic change in immunomodulatory effect resulting in the modulation of innate and adaptive immune responses in mice. Hence, it would be suggest that bengkoang may have beneficial effect on prevention on human health of immune-system related diseases.

**Author Contributions:** Conceptualization, A.E.N., E.L. and A.N.; data curation, H.N.B. and A.N.; formal analysis, H.N.B., A.E.N., E.L. and A.N.; funding acquisition, A.N.; investigation, H.N.B. and A.N.; methodology, H.N.B., A.E.N., E.L. and A.N.; project administration, H.N.B. and A.N.; resources, A.N.; software, A.N.; supervision, A.E.N., E.L. and A.N.; validation, H.N.B., A.E.N., E.L. and A.N.; writing—original draft, H.N.B. and A.N.; writing—review and editing, H.N.B., A.E.N., E.L. and A.N. All authors have read and agreed to the published version of the manuscript.

**Funding:** This research was funded by the Ministry of Research Technology and Higher Education of the Republic of Indonesia, International Research Collaboration and Scientific Publication, under Grant No. 1672/UN1/DITLIT/DIT-LIT/LT/2018.

**Acknowledgments:** The authors thank Takuya Sugahara from Ehime University for collaborating in this research.

**Conflicts of Interest:** The authors declare no conflict of interest. The funders had no role in the design of the study; in the collection, analyses, or interpretation of data; in the writing of the manuscript, or in the decision to publish the results.

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
