# Peer review of "Water-Soluble Fiber from Bengkoang (Pachyrhizus erosus (L.) Urban) Tuber Modulates Immune System Activity in Male Mice"

_scipharm, doi:10.3390/scipharm88030034_

Round 1
Reviewer 1 Report
In this paper by Baroroh et al, the Authors investigated the immunomodulatory effect of two fractions of water soluble fiber extracts from bengkoang in mice. Animals were treated with hepatitis B vaccine and then they were fed with two different concentrations of the extracts. Results showed that extracts differently modulate immune system, NO synthesis and cytokines production.
The study is quite elegant, the design is appropriate and the methodologies used are pertinent. Despite these considerations some issues raised.
Major revision:
- The Authors should chemically characterize the fiber extracts.
- Discussion section reported mainly similar or contrasting results present in the current literature with the results obtained without given a putative explanation on the mechanism of action.
Minor revision:
- Authors could perhaps include a figure when describing the Material and Methods with the experimental protocol design. This will help readers to better follow the sequence in which the treatments were performed.
- Explanation of the effects of levamisole should be moved from discussion (line 277-279) to paragraph 2.5
- Figure 1A and B should be incorporated in only one figure. Same for 4A and B, 4C and D, and figure 5.
- Check the English by a mother-tongue
Author Response
Response to Reviewer 1 Comments
Major Revision:
Point 1: The Authors should chemically characterize the fiber extracts.
Response 1: Thanks for your comment.
Chemical characterization of fiber fractions has been reported by Baroroh et al (2020) in our previous studies. *(Baroroh, H.N.; Nugroho, A.E.; Lukitaningsih, E.; Nurochmad, A. Immune‑enhancing Effect of bengkoang (Pachyrhizus erosus (L.) Urban) fiber fractions on mouse peritoneal macrophages, lymphocytes, and cytokines. Journal of Natural Science, Biology and Medicine. (accepted)
We also have improved the discussion. The characterization of fiber fractions was cited in the introduction and discussion.
Introduction: page number 2, line 56-59 (Ref.11)
Discussion: page number 10, line 327-328 Ref.11)
Point 2: Discussion section reported mainly similar or contrasting results present in the current literature with the results obtained without given a putative explanation on the mechanism of action.
Response 2: Thanks for this comment. We have improved the discussion.
Minor revision:
Point 1: Authors could perhaps include a figure when describing the Material and Methods with the experimental protocol design. This will help readers to better follow the sequence in which the treatments were performed.
Response 1: We have changed and improved experimental design with a figure.
Section 2.4: page number 3, line 102-103 (Figure 1)
Section 2.5: page number 4, line 119-120 (Figure 2)
Point 2: Explanation of the effects of levamisole should be moved from discussion (line 277-279) to paragraph 2.5
Response 2: We have changed and moved to section 2.5 (experimental animals).
Page number 3, line 111-114
Point 3: Figure 1A and B should be incorporated in only one figure. Same for 4A and B, 4C and D, and figure 5.
Response 3: We have changed and improved figures:
Figure 1A and B to be Figure 3: page number 6, line 190-195 (Figure 3)
Figure 4A and B, 4C and D to be Figure 6: page number 8, line 251-255 (Figure 6)
Figure 5 to be Figure 7: page number 8, line 261-264 (Figure 7)
Point 4: Check the English by a mother-tongue
Response 4: Thanks for your valuable comment and suggestion.
The manuscript was submitted and professionally edited by a professional English Language editing agency. We have changed and improved a few errors in grammar and style.
Reviewer 2 Report
The paper Water-soluble fiber from bengkoang (Pachyrhizus erosus (L.) Urban) tuber modulates immune system activity in male mice deals with imunomodulatory effects of bengkoang tuber water extract and fiber fraction B.
Experiments are described in a well manner as well as results. Conclusion is supported with results.
Discussion is a little bit too long - perhaps it could be a bit shortened.
Few errors in grammar and style are indicated in .pdf file.
Also authors have referred to their own article accepted for publication in J. Nat. Sci. Biol. Med. titled:Immune‑enhancing Effect f 398
Bengkoang (Pachyrhizus erosus (l.) Urban) Fiber Fractions on Mouse Peritoneal Macrophages, Lymphocytes, and Cytokines.
NOTE: If the same fiber fraction B was tested in this paper as well - then it should be clearly cited in abstract and introduction.
Otherwise I cannot see any obstacle in accepting this paper for publication.

Author Response
Response to Reviewer 2 Comments
Point 1: Discussion is a little bit too long - perhaps it could be a bit shortened.
Response 1: Thanks for your comment.
We have deleted some sentences in discussion and/or moved to another section.
Point 2: Few errors in grammar and style are indicated in .pdf file.
Response 2: Thanks for your comment.
We have changed and improved a few errors in grammar and style:
Page number 3, line 97
Page number 4, line 143-144
Page number 9, line 274-279
Page number 9, line 301
Page number 11 line 384
Citation for the sentence "BFE could activate J774.1 cells through the MAPK and NFkB pathways [8], enhance the phagocytic activity of macrophages, and stimulate the production of TNF and IL-6 [9] is not mistaken, because it actually uses the same cell line (J774.1 cells).
Page number: 2 line : 53-55
Point 3: Also authors have referred to their own article accepted for publication in J. Nat. Sci. Biol. Med. titled:Immune enhancing Effect f 398
Bengkoang (Pachyrhizus erosus (L.) Urban) Fiber Fractions on Mouse Peritoneal Macrophages, Lymphocytes, and Cytokines
NOTE: If the same fiber fraction B was tested in this paper as well - then it should be clearly cited in abstract and introduction.
Response 3: We have cited in the abstract and introduction.
Abstract: page number 1, line 16-18
Introduction: page number 2, line 56-59 (Ref.11)
Round 2
Reviewer 1 Report
The revised version of the paper which includes tracking of all modifications it’s hard to read and understand. Please provide a new revised version with only the new parts of the text/figure highlighted or in red. Do not include the text cuts out.
Author Response
Response to Reviewer 1 Comments
Major Revision:
Point 1: The Authors should chemically characterize the fiber extracts.

Response 1: Thanks for your comment.
Chemical characterization of fiber fractions has been reported by Baroroh et al (2020) in our previous studies. *(Baroroh, H.N.; Nugroho, A.E.; Lukitaningsih, E.; Nurochmad, A. Immune‑enhancing Effect of bengkoang (Pachyrhizus erosus (L.) Urban) fiber fractions on mouse peritoneal macrophages, lymphocytes, and cytokines. Journal of Natural Science, Biology and Medicine. (accepted)
We also have improved the discussion. The characterization of fiber fractions was cited in the introduction and discussion.
Introduction: page number 2, line 56-59 (Ref.11)
Discussion: page number 10, line 329-330 Ref.11)
Point 2: Discussion section reported mainly similar or contrasting results present in the current literature with the results obtained without given a putative explanation on the mechanism of action.
Response 2: Thanks for this comment. We have improved the discussion and highlighted some sentences that may give a putative explanation of the mechanism of action.
Minor revision:
Point 1: Authors could perhaps include a figure when describing the Material and Methods with the experimental protocol design. This will help readers to better follow the sequence in which the treatments were performed.
Response 1: We have changed and improved experimental design with a figure.
Section 2.4: page number 3, line 102-103 (Figure 1)
Section 2.5: page number 4, line 119-120 (Figure 2)
Point 2: Explanation of the effects of levamisole should be moved from discussion (line 277-279) to paragraph 2.5
Response 2: We have changed and moved to section 2.5 (experimental animals).
Page number 3, line 112-114
Point 3: Figure 1A and B should be incorporated in only one figure. Same for 4A and B, 4C and D, and figure 5.
Response 3: We have changed and improved figures:
Figure 1A and B to be Figure 3: page number 6, line 190-191 (Figure 3)
Figure 4A and B, 4C and D to be Figure 6: Page number 8, line 251-252 (Figure 6)
Figure 5 to be Figure 7: page number 8, line 261-262 (Figure 7)
Point 4: Check the English by a mother-tongue
Response 4: Thanks for your valuable comment and suggestion.
The manuscript was submitted and professionally edited by a professional English Language editing agency. We have changed and improved a few errors in grammar and style.
Round 3
Reviewer 1 Report
No further comments